# A Design and Implementation Using an Innovative Deep-Learning Algorithm for Garbage Segregation

**DOI:** 10.3390/s23187963

**Published:** 2023-09-18

**Authors:** Jenilasree Gunaseelan, Sujatha Sundaram, Bhuvaneswari Mariyappan

**Affiliations:** 1Department of Computer Applications, University College of Engineering, Anna University (BIT Campus), Trichy 620 024, Tamilnadu, India; sujathaaut@gmail.com; 2Department of ECE, University College of Engineering, Anna University (BIT Campus), Trichy 620 024, Tamilnadu, India

**Keywords:** ResNeXt, convolution neural network, classification, smart garbage collector, ultrasonic sensor, temperature sensor, solar panel

## Abstract

A startling shift in waste composition has been brought on by a dramatic change in lifestyle, the quick expansion of consumerism brought on by fierce competition among producers of consumer goods, and revolutionary advances in the packaging sector. The overflow or overspill of garbage from the bins causes poison to the soil, and the total obliteration of waste generated in the area or city is unknown. It is challenging to pinpoint with accuracy the specific sort of garbage waste; predictive image classification is lagging, and the existing approach takes longer to identify the specific garbage. To overcome this problem, image classification is carried out using a modified ResNeXt model. By adding a new block known as the “horizontal and vertical block,” the proposed ResNeXt architecture expands on the ResNet architecture. Each parallel branch of the block has its own unique collection of convolutional layers. Before moving on to the next layer, these branches are concatenated together. The block’s main goal is to expand the network’s capacity without considerably raising the number of parameters. ResNeXt is able to capture a wider variety of features in the input image by using parallel branches with various filter sizes, which improves performance on image classification. Some extra dense and dropout layers have been added to the standard ResNeXt model to improve performance. In order to increase the effectiveness of the network connections and decrease the total size of the model, the model is pruned to make it smaller. The overall architecture is trained and tested using garbage images. The convolution neural Network is connected with a modified ResNeXt that is trained using images of metal, trash, and biodegradable, and ResNet 50 is trained using images of non-biodegradable, glass, and hazardous images in a parallel way. An input image is fed to the architecture, and the image classification is achieved simultaneously to identify the exact garbage within a short time with an accuracy of 98%. The achieved results of the suggested method are demonstrated to be superior to those of the deep learning models already in use when compared to a variety of existing deep learning models. The proposed model is implemented into the hardware by designing a three-component smart bin system. It has three separate bins; it collects biodegradable, non-biodegradable, and hazardous waste separately. The smart bin has an ultrasonic sensor to detect the level of the bin, a poisonous gas sensor, a stepper motor to open the lid of the bin, a solar panel for battery storage, a Raspberry Pi camera, and a Raspberry Pi board. The levels of the bin are maintained in a centralized system for future analysis processes. The architecture used in the proposed smart bin properly disposes of the mixed garbage waste in an eco-friendly manner and recovers as much wealth as possible. It also reduces manpower, saves time, ensures proper collection of garbage from the bins, and helps attain a clean environment. The model boosts performance to predict waste generation and classify it with an increased 98.9% accuracy, which is more than the existing system.

## 1. Introduction

As urbanization and population growth continue to increase, so does the production of waste, leading to various environmental and public health challenges. To address this issue, metropolitan governments must implement comprehensive waste management strategies that focus on reducing, reusing, and recycling waste while ensuring proper disposal and treatment of the remaining waste. Encouraging individuals and businesses to take ownership of their waste generation and adopt sustainable practices can make a significant difference. Implementing waste segregation: Promoting waste segregation at the source is essential for effective waste management. This involves separating different types of waste (organic, recyclable, non-recyclable) at the point of origin, making it easier to recycle and treat waste appropriately. Develop comprehensive recycling programs: Establishing and supporting recycling programs is crucial to diverting waste from landfills and reducing the environmental impact of waste disposal. Invest in waste treatment and disposal facilities: Metropolitan governments should invest in modern waste treatment and disposal facilities, including composting plants, recycling centers, waste-to-energy facilities, and sanitary landfills. These facilities can help manage different types of waste more efficiently and minimize the adverse effects on the environment. Promote extended producer responsibility (EPR): EPR policies make manufacturers responsible for the end-of-life disposal of their products and packaging [1,2,3]. Implementing EPR could encourage manufacturers to design more eco-friendly products and packaging, leading to reduced waste generation. Encourage circular economy practices: Metropolitan governments should promote a circular economy approach, where waste materials are seen as resources rather than discarded items. This involves designing products for durability, reparability, and recycling and encouraging the use of recycled materials in manufacturing. Monitoring the performance of waste management initiatives helps identify areas of improvement and ensures that the strategies are effective. Implement smart technologies: Utilizing smart technologies [4,5], IOT-based waste monitoring systems and data analytics, could optimize waste collection routes and enhance overall waste management efficiency. The current manual methods of waste separation and categorization can be labor-intensive, time-consuming, and costly. Automated sorting systems offer promising alternatives to streamline waste management processes and improve overall efficiency. Across six classes (paper, glass, plastic, metal, cardboard, and trash), The dataset [6] (TrashNet dataset) contains photographs of recycled materials with an average of 500 images per class. The testing accuracy was 63% and 27%, respectively, for SVM and TrashNet, which have an architecture similar to AlexNet but with fewer and smaller filters. The TrashNet dataset was chosen and tested it with a variety of deep CNN architectures and optimization techniques [7,8]. By changing the connection patterns of the skip connections inside dense blocks, they aim to shorten prediction times based on the existing models. With a test accuracy of 95% as a result, DenseNet121 with transfer learning obtained the best performance. Similar results were obtained [9]. These systems leverage technology to identify and classify different forms of waste. The authors had suggested the creation of an IoT-based waste management system; however, the authors failed to offer a structural design of the garbage management system utilizing the deep learning paradigm. The rubbish collection robot that could move on the ground was introduced by the paper’s author [10]. The proposed architecture, according to the authors, used deep learning algorithms to accurately detect junk [11]. The prototype had a waste detection accuracy of 95%. The author of the research has suggested a creative floor-cleaning robot. The trade-off between a tile’s area coverage and energy consumption was determined by the proposed model, which used a fuzzy inference approach [12,13]. The concept allowed for the weighted sum concept (WSM), which is based on a fuzzy inference system’s representation of user preferences for multiple criterion decision-making (MCDM). The authors of the study papers introduced a prototype of a waste management robot employing deep learning and fuzzy inference systems, respectively, but they did not acknowledge their contributions to the Internet of Things.

Machine learning (ML) and artificial intelligence (AI) play a crucial role in modern waste management efforts. ML enables systems to learn and make decisions based on data without being explicitly programmed. It allows waste management systems to analyze and process large amounts of data related to waste generation, disposal patterns, recycling rates, and other relevant factors [14,15,16]. By learning from this data, ML algorithms can optimize waste collection routes, predict fill levels in garbage bins, and identify patterns to improve overall waste management efficiency.

Deep learning is a subset of ML that focuses on training complex neural networks with multiple layers to recognize patterns and features in data. CNNs (“Convolutional Neural Networks”) are a specific form of DL (“Deep Learning”) architecture that excels in image recognition tasks. In the context of waste management, CNNs could be utilized for various purposes:Image recognition: CNNs could be employed to identify and categorize waste items from images, facilitating automated sorting and recycling processes [17].Visual inspection: CNNs can be integrated into waste separation systems to visually inspect waste streams and separate recyclable materials from non-recyclables [18].Waste characterization: CNNs can assist in identifying hazardous waste or specific waste types that require specialized handling and disposal methods.Anomaly detection: CNNs can be used to detect irregularities in waste streams, such as prohibited items or contaminants, improving the overall quality of recycling processes. In the existing system, there are automated waste sorting systems that can employ various technologies, such as sensors, cameras, and artificial intelligence. However, there is a lack in this existing method; the garbage is not classified properly, and it requires more time to classify it [19]. It can be concluded that most of the current automated waste management systems using various hardware cannot identify and classify the image correctly, and it requires more time to classify it.

To overcome this issue, we proposed a novel approach by combining two technologies, namely modified ResNeXt and ResNet 50, with IoT to create an efficient garbage management solution [20,21]. The modified innovative ResNeXt Model and ResNet 50 are combined as a multilayer and are responsible for image classifying of various garbage accurately in less time. The proposed ResNeXt model is an extension of the existing ResNeXt model that consists of a convolution neural network and pooling layer. It consists of vertical and horizontal blocks, which lead to the easiest way of image classification. For real-time practice, we have created three compactor smart bins, and the smart bin is divided into three phases: biodegradable, non-biodegradable, and hazardous. The proposed ResNeXt model and ResNet 50 are combined in a parallel way, and for real-time application, it is applied to the hardware of the smart bin [22,23]. By leveraging IoT, deep learning, and sensor technologies, the proposed smart trash can autonomously identify and sort waste, monitor fill levels, and send real-time data to a centralized platform for further analysis and decision-making. This kind of intelligent waste management solution has the potential to significantly enhance efficiency, decrease environmental effects, and contribute to a more sustainable and cleaner urban environment. In a smart bin, the waste images associated with other numerical information measured by the sensor are fed into the system [23,24,25,26,27,28]. The smart bin model is designed using an ultrasonic sensor to identify the amount of garbage filled in the bin, an LED light to indicate the level of the bin, an LCD screen to display the levels, a GSM module for sending the information to the particular website, and a mobile phone for the collection of garbage from the bins [29,30]. The data stored on the website is used for analysis and prediction processes to avoid improper collection of garbage. The smart bin helps with the integration into waste sorting facilities to automatically detect and separate recyclable materials, organic waste, and non-recyclable items. By automating the sorting process, the need for extensive manual labor is reduced, making waste management more cost-effective and less manpower demanding [31,32].

The suggested solution employs a clever approach that allows users to take essential safety precautions regarding the waste management system. The following are the paper’s main contributions: An innovative technique to integrate two technologically modified ResNeXt and ResNet 50, which is implemented in the IoT and deep learning, guarantees an ideal solution in the field of waste management.Using deep learning to identify images is a clever method of separating biodegradable waste from non-biodegradable rubbish and hazardous waste.A poisonous gas sensor, stepper motor, solar panel, load measurement sensor, and ultrasonic sensor are used in the architectural design of a smart garbage can.Using an application and Bluetooth for short-range and the Internet of Things for long-range monitoring of waste, respectively, the data is stored for further analysis.

The following are the paper’s contributions: The paper presents several contributions related to garbage management and waste classification using IoT (Internet of Things) and deep learning techniques.

## 2. Related Works

In this field, a number of research articles were already published. Attention is being paid to this area increasingly due to its significance in maintaining ecological balance and promoting sustainable development. Table 1 describes the related works of various papers.

The authors of [40,41] presented a deep learning model and Internet of Things-based smart bins that employ a microcontroller and sensors to monitor waste in real-time and separate it into digestible and indigestible waste using a CNN. Machine learning is used to classify indigestible materials into six different categories, with an accuracy of up to 95%. The operation of the camera and sensor is shown, along with a thorough explanation of how a smart bin operates. The plan is suggested for implementation for both the segregation of household garbage and real-time monitoring. It is obvious that using a variety of sensors, such as MQ Gas sensors and IR sensors, would help to increase accuracy [42].

The model discussed in [43] takes into account a TrashNet dataset that is examined for the classification of garbage images. With the aid of a multilayer hybrid convolution neural network, a straightforward, organized, and effective trash categorization system with fewer parameters has been proposed, with an accuracy of up to 92%. In order to examine the best classification and accuracy, the basic model has been proposed with four subsequent upgraded versions back-to-back. In order to extract rich features, a suitable heat map comparison in the form of photographs is also provided. The suggested technique has also been compared to existing ones in terms of accuracy, parameters, and complexity.

In [44,45] the authors suggest utilizing binary classification to identify empty recycling bins in order to use machine learning to address waste management concerns. Using the rigorously applied data pretreatment approaches in this approach, the model outperforms the previous frameworks. In contrast to conventional approaches without feature engineering, it generalizes the model using a feature engineering methodology, increasing accuracy. On the three fundamental approaches—the feature engineering model, conventional machine learning with standard features, and extended features—performance optimization was carried out.

A clever mechanism for the forecasting and prediction of waste output is put forth in [46]. Data preparation and integration are performed using CNN and an air jet structure. The model’s outcomes demonstrate that the suggested method is highly effective at accurately identifying various waste categories. The program primarily intends to address waste-related concerns in the well-known Indian city of Dehradun. Both CNN and decision-tree algorithms can be used to quickly identify and separate junk using the developed model [47]. The technology allows for automatic trash identification with minimal human involvement, improving disease and pollution control [48].

## 3. Dataset

This system employs a Thung G [made garbage dataset] and includes some more photos for our analysis. The dataset is divided into seven categories: everyday garbage, glass, cardboard, paper, plastic, metal, and trash. There are 10,314 photos in this collection, which are distributed as follows: 1500 photographs of plastic, 1859 images of paper, 1600 images of glass, 1405 images of cardboard, 1300 images of metal, and 500 images of garbage. In this dataset, we also included extra images of various garbage as well as the image of bio-degradable added into it, and Figure 1 represents the sample images of various garbage and Table 2 describes about dataset information.

## 4. Proposed Methodology

The methodology proposed in the paper combines two important components to enhance waste management: the creation of smart trash bins using convolutional neural networks (CNNs) for waste classification and the incorporation of IoT-enabled real-time data monitoring. By combining these two structural models, the model will achieve outstanding outcomes in the field of waste management. The first component involves using CNNs, which are a type of deep learning algorithm known for their exceptional performance in image classification tasks. The researchers utilize deep learning algorithms to develop a waste classification system that accurately detects different garbage forms, including biodegradable, non-biodegradable, and hazardous waste. Properly classifying waste is crucial because it allows the system to identify recyclable items, enabling their reuse instead of letting them degrade or become waste.

The input characteristics are first sent to the channel attention module. The tensor all-channel feature matrix is computed first, followed by channel attention, which is efficiently computed by utilizing global average and maximum pooling to assemble two matrices. Weight optimization training supplies these two weight matrices into the same multiple-layer perceptron. The following two output components are combined to create a channel weighting module shown in Figure 2a. The data is compressed using the sigmoid activation function to add the input features and multiply it by a value between 0 and 1, where AvgPool and MaxPool stand for the global average pooling and global maximum pooling, respectively, and F stands for the input features. ReLU serves as the first layer’s activation function in a multilayer perceptron (MLP), which has two completely connected layers that stand for the sigmoid activation function.

After going via the channel attention module, the feature maps arrive at the spatial attention module. The spatial attention module processes the input using average pooling and maximum pooling, although compressed sampling is carried out in the channel dimension. Two two-dimensional spatial matrices are created after the pooling process, concatenated in the channel dimension, followed by a convolutional layer. In this work, each of the underlying residual blocks of ResNeXt has the CBAM module. The standard ResNeXt model should be used when down sampling is necessary. It utilizes a 2-step size 11 convolutional kernel to perform the convolutional procedure, which always causes the loss of information. Consequently, convolutional procedures use a step and a convolutional kernel of size 1 × 1. In this post, a size of 2 will not be used shown in Figure 2b. The above block shows the new base block structure, the initial framework. Table 3 describes the accuracy shown.

Table 4 shows the comparison of the proposed model. The second component focuses on real-time data monitoring, made possible by integrating the Internet of Things (IoT) technology into the smart trash bins. Each smart bin is equipped with various sensors, including ultrasonic and thermal sensors, a GSM module for data communication, an LCD display for local information, a Pi camera for image input, and an LED display for output. The Pi camera captures an image of the garbage inside the bin as input, and then deep learning approaches are used to identify the waste. Depending on the classification result, the corresponding smart bin light will glow, indicating the type of waste present. Figure 3 of the paper illustrates the Pi camera workflow and the classification process, which illustrates how the image is captured and processed through the deep learning model, and the output is used to trigger the LED display. By combining these technologies and methodologies, the proposed smart trash bins can significantly improve waste management efficiency. They can autonomously identify recyclable items, reduce contamination of recyclables, and provide real-time data insights for waste management decision-making. This integrated approach has the potential to revolutionize waste management practices and contribute to a more sustainable and environmentally friendly waste disposal system.

The ultrasonic sensor working flow in the smart garbage bin is as follows:Initialization: The microcontroller initializes the ultrasonic sensor by sending a high-to-low signal at the trigger pin. This signal triggers the ultrasonic sensor to start generating sound waves.Sound wave generation: After receiving the trigger signal, the ultrasonic sensor generates eight 40 kHz sound waves. These sound waves travel from the sensor outward into the garbage bin.Echo detection: The ECHO pin of the ultrasonic sensor is turned high and continuously monitored. When the sound waves encounter an object, such as garbage, inside the smart bin, they bounce back (echo) to the sensor.Time measurement: The microcontroller uses Timer1 to measure the time taken for the sound waves to echo back to the sensor. This time duration is recorded for further calculations.Distance calculation: The calculated time duration is then used in the following equation to determine the distance (D) between the ultrasonic sensor and the object (garbage) in the smart bin:
D = (t × v)/2
where:

D = Distance in centimeters (cm) determined by the ultrasonic sensor.

t = Time taken by the sound wave to echo back to the receiver in seconds.

v = Sound wave velocity.

By knowing the speed of sound in the medium (usually air), which is approximately 343 m per second (m/s) at room temperature, the distance can be calculated. The division by 2 in the equation is because the sound wave travels from the sensor to the object and then back, so the total distance is twice the distance from the sensor to the object. This process allows the ultrasonic sensor to track how full the smart garbage bin is as it is being filled showed in Figure 4. By continuously monitoring the distance between the sensor and the garbage at different levels, the system can estimate the fill level of the bin and trigger appropriate actions, such as signaling when the bin is approaching full capacity or needs to be emptied. Additionally, the thermal sensor in the smart bin is used to identify whether a fire exists inside the bin. It can detect unusual temperature increases, indicating a potential fire hazard, allowing for timely intervention and prevention of further damage.

The described system uses sensors in the smart garbage bins to monitor the fill level every time waste crosses the sensors. This data is then sent via the GSM module as instant messages to the garbage analyzer. The microcontroller in the smart bin computes the distance using the previously mentioned equation, allowing for the distance determination of any solid waste material present in the bin. At the garbage analyzer end, each message received from the smart bins is saved as data, which is then utilized for analysis and predictive modeling. The application interface utilizes real-time data to display the filled level of the smart bins. The data is sent as text messages to the App, which acts as a data warehouse, storing all the level data sent by different bins. This centralized data storage enables the waste management department to track the fill level of each dustbin in real-time from their office. Table 5 likely represents the total cost associated with implementing this smart garbage management system. The cost may include the expenses related to the hardware components, such as the Raspberry Pi board, sensors (ultrasonic, thermal), GSM module, LCD display, and Pi camera. Additionally, the cost might cover the software development, data analysis tools, and communication infrastructure required for data transmission.

By using this integrated system of sensors, GSM communication, and data analysis, waste management authorities can have a better understanding of the fill levels of garbage bins, enabling more efficient waste collection routes and decreasing the risk of overflowing bins. This approach can lead to improved waste management practices, reduced operational costs, and a cleaner environment for urban areas.

The database plays a crucial role in storing the received data from the smart garbage bins while preserving all of its properties, including the time and date when the data was collected. This historical data is then utilized by the data analysis department for various purposes, such as making predictions about waste generation trends and generating insightful reports. By storing data over months, the system can establish patterns and trends in waste generation, enabling the waste management authorities to make informed decisions and optimize waste collection schedules. This data-driven approach can lead to more efficient resource allocation, reduced operational costs, and improved overall waste management strategies. The hardware components used in the smart garbage bins are designed with cost-effectiveness in mind. By utilizing basic electronic components, the system could be implemented in any dustbin, regardless of its size and height. This portability ensures that the smart garbage bin can be deployed in various locations without significant modifications, making it a flexible and scalable solution.

Figure 5 represents the 3D view of the proposed smart garbage bin, providing a visual representation of the hardware’s design. This view showcases the various components, such as the Raspberry Pi board, sensors, GSM module, LCD, Pi camera, and LED display, integrated into the smart bin to enable its functionalities. Overall, the combination of real-time data collection, historical data analysis, and cost-effective hardware design makes this smart garbage management system a powerful tool for waste management authorities. By leveraging technology and data-driven insights, this system can contribute to more efficient waste management, reduced environmental impact, and a cleaner, more sustainable urban environment.

### Benefits of Proposed Smart Bin

The proposed smart bin has numerous features and benefits that make it an advanced and efficient waste management solution than the existing system [49]. The bin uses the proposed architecture for image classification. The suggested methodology combines two components: the creation of smart trash cans using convolutional neural networks for waste classification and IOT-enabled real-time data monitoring. To achieve outstanding outcomes in waste management, two structural models are combined. Garbage that can be reused can be identified by properly classifying it. Recognizing recyclable wastes enables us to use them without having them degrade. Deep learning algorithms provide unmatched performance in the area of picture classification. The proposed algorithm classifies the image correctly within a short period; even though the image is blurred or crushed, it is classified correctly.

Here are the key features and advantages of the proposed system:High capacity: The smart bin has a large capacity of 600 L, which is five times the capacity of typical 120 L bins. This high capacity reduces the frequency of waste collection rounds, saving time and resources.Automatic compaction: The smart bin is equipped with an internal compactor that automatically compresses the waste when it is full. This increases the trash capacity by 6 to 8 times that of a regular street bin, reducing the need for frequent emptying.Industry-leading hopper design: The hopper design of the smart bin is designed to keep waste confined, preventing insect access, waste overflow, and wind-blown litter. This helps maintain cleanliness and hygiene in public spaces.Significant waste reduction: The smart bin’s capacity and automatic compaction result in an average 86% reduction in street waste collected. This reduction in waste collection helps in efficient waste management and resource optimization.Real-time fill level monitoring: The smart bin is equipped with a sensing system that sends an email and text notification when the bin reaches 85% fullness. This real-time monitoring allows management and workers to plan waste collection rounds more effectively.Remote monitoring: Management and workers can observe the real-time fill levels of all smart bins on desktops and smartphones. This feature eliminates the need for traditional “milk round” type collection rounds, making waste collection more data-driven and efficient.Compostable-friendly: The smart bin is designed to support compostable waste collection with timely pickups and an enclosed design. This helps promote sustainable waste management practices.Elimination of overflowing bins: With the internal compactor and real-time monitoring, the smart bin effectively eliminates overflowing bins, preventing littering and maintaining cleanliness in public spaces.No electricity required: The smart bin does not require electricity for fullness-level sensing or communication. This makes it more energy-efficient and suitable for various locations, including areas with limited power resources.Smart bin security system: The smart bin is equipped with a security system that includes physical security plates to bar unauthorized access and a security management module to set up and manage security events effectively.

The proposed smart bin is able to achieve excellent accuracy in real-time. The overall accuracy is higher than 99.5% compared to the existing method.

## 5. Hardware Required

### 5.1. Raspberry Pi Board

The Raspberry Pi has gained immense popularity due to its low cost, compact size, and ease of use. It has become a valuable tool for hobbyists, educators, and professionals alike, allowing them to create a wide range of projects and applications.

Increased memory: It comes with 1 GB of RAM, providing more memory for running complex applications and handling larger data sets.Built-in wireless connectivity: The Raspberry Pi 3B has built-in Wi-Fi and Bluetooth capabilities, making it easier to connect to the internet and various wireless devices.Multiple USB ports: It has four USB ports, allowing users to connect various peripherals, such as keyboards, mice, external hard drives, and more.Ethernet port: The Raspberry Pi 3B includes an Ethernet port for wired network connectivity.HDMI output: It has an HDMI port for connecting to a monitor or TV, enabling users to use it as a desktop computer.GPIO pins: The Raspberry Pi 3B retains the GPIO (general purpose input/output) pins, which allow users to interface with various electronic components and sensors for electronic projects.

### 5.2. Ultra Sonic Sensor and Temperature Sensor

The oscillator receives and emits ultrasonic waves alternately, and it is highly effective at measuring amplitudes. This enables the oscillator to be an ultrasonic sensor for both emission and reception. It allows it to miniaturize the sensor. The distance is determined using:L = 12 × T × C.
where T stands for the elapsed time between emission and reception, C is referred to as the sonic speed, and L is the distance. Using an ultrasonic sensor, it is possible to correctly determine each trash container’s fill level in real time (HC-SR04). Using an ultrasonic transmitter and receiver, it can measure an object’s distance from 2 cm to 400 cm. Moreover, signaling with an LED light effectively lowers the garbage overflow. The use of LED lights to indicate the fill level of the trash bin is a simple and effective way to provide visual feedback to users and waste management personnel.

### 5.3. LCD Display and Solar Panel

The output module in the smart trash bin incorporates a 16 × 2 LCD display system to provide visual feedback on the garbage fill level. The LCD display can print up to 32 characters at once, allowing it to show relevant information about the trash bin’s status. The ultrasonic sensor, which is placed inside the bin, continuously measures the level of rubbish present. As the garbage fills up in the bin, the ultrasonic sensor calculates the distance from the top of the garbage to the sensor, using the time taken for the sound waves to echo back. This data is then processed by the microcontroller to estimate the fill level of the bin. The estimated fill level is then displayed on the 16 × 2 LCD display. The LCD can show up to 32 characters at a time, providing a clear and concise representation of the garbage fill level. This real-time display of garbage fill level on the LCD helps users and waste management personnel monitor the bin’s status without the need for manual inspection.

### 5.4. Raspberry PI Camera

The Pi camera is a versatile and affordable camera module that can be directly connected to the Raspberry Pi via the CSI (Camera Serial Interface) connector. This interface enables high data transfer speeds between the camera and the Raspberry Pi, making it suitable for various applications that require real-time video processing. One of the key selling points of the Pi camera is its accessibility and low cost, making it widely popular among Raspberry Pi users. It supports various video resolutions, including 1080p at 30 frames/s, 720p at 60 frames/s, as well as 640 × 480p at 60/90 frames/s. This flexibility allows users to choose the appropriate resolution for their specific application requirements. With its compact dimensions (20 × 25 × 9 mm) and lightweight design (3 g), the Pi camera is portable and can be easily integrated into various Raspberry Pi cases without adding significant bulk or weight.

## 6. Experimental Results

The proposed model in the study accurately forecasts the given input image with less valuation loss compared to previous models. Figure 6 depicts the connection of the waste bin to the Raspberry Pi board, indicating the integration of the hardware-based waste segregation system.

The model’s accuracy can be significantly improved if the dataset used for training is larger. However, the researchers encountered difficulties due to the vast range of differences between diverse objects. For example, a single glass item can come in various colors and thicknesses, making accurate classification more challenging. In this study, the waste was divided into seven categories: cardboard, metal, glass, paper, plastic, biodegradable waste, and garbage. Despite achieving 98.9% accuracy, the model performs considerably better in practice because this accuracy refers to categorizing garbage into seven groups. The real-time analysis of the garbage, presenting the results of the proposed system’s garbage classifier.

In Figure 7 and Table 6 shows the confusion matrix is displayed and comparison of various algorithm of confusion matrix. Class-0 stands for the MSI, and Class-1 for the MSS. When MSI is anticipated, and the actual output is MSI, this is referred to as a true positive (TP). When MSS is anticipated, and the actual output is MSS, the condition is referred to as a true negative (TN). Similar to the case where MSI is anticipated but the actual output is MSS, this is known as a false positive (FP), and when MSS is predicted, but the actual result is MSI, this is known as a false negative (FN).

The classifier was trained using a dataset with six classes, and if the waste is mixed, the system is still capable of appropriately classifying it and placing it in the correct trash receptacle. Even with two types of garbage cans for waste collection, the model is capable of accurately categorizing metal as glass or plastic, and vice versa, in the overall scenario shown in Figure 8. According to the confusion matrix statistics, the accuracy of proper box classification is reported to be 98.9%. The overall findings suggest that the proposed hardware-based waste segregation system, powered by a trained CNN model, can effectively and accurately categorize waste items into appropriate groups. The model’s high accuracy in practical scenarios enhances waste management efforts, enables more efficient recycling practices, and contributes to a cleaner and more sustainable environment.

Overall, the smart bin’s features and benefits contribute to a more efficient, clean, and environmentally friendly waste management system in public spaces, improving the quality of urban environments and promoting sustainable waste disposal practices. Table 7 describes about the real time analysis of the proposed bin. 

## 7. Conclusions

The research presented here describes a real-time trash monitoring system that makes use of the IoT and deep learning paradigms. In order to ensure an effective waste management procedure, the research is carried out using a set of development processes. The proposed model is divided into two key components. One uses a Raspberry Pi, a camera module, and deep learning as part of an architectural model for classifying waste. Another example is the implementation of an Internet of Things-based smart garbage can that makes use of a Raspberry Pi and many sensors to monitor waste in real-time. Once more, the approach of the suggested CNN model, ultrasonic sensor, and load measurement sensor in this study is modified by ResNeXt and ResNet 50 and data analysis to achieve efficient waste segregation. The process begins with object classification through a CNN. The researchers use image classification techniques to differentiate waste types. The CNN is trained using waste images to accurately categorize different types of waste. This approach enables the system to identify eight waste categories: bio-degradable, non-biodegradable, hazardous waste, plastic, metal, paper, glass, and cardboard. The real-time embedded system is implemented using a Raspberry Pi board, which acts as the core processing unit. This system continuously captures and analyzes waste images from the smart garbage bins using the trained CNN model. The real-time data from the smart bins is stored in the Internet of Things (IoT), allowing for seamless communication and monitoring. Data analysis is a critical aspect of the system, which uses stored real-time data to generate valuable insights and predictive models. By analyzing historical waste data, waste management authorities can make informed decisions, optimize waste collection routes, and plan timely waste disposal strategies. The hardware-based waste segregation system provides an efficient and automated solution for waste categorization. By combining object classification through CNN, real-time data monitoring using Raspberry Pi and IoT, and data analysis, the system can accurately segregate waste into appropriate categories. This approach not only simplifies waste management processes but also promotes sustainable waste disposal practices, recycling efforts, and environmental conservation. The system’s ability to categorize waste into eight distinct types enables more effective recycling initiatives and ensures that each waste item is disposed of appropriately, minimizing the environmental impact of waste accumulation. Overall, this hardware-based waste segregation approach has the potential to significantly enhance waste management practices, contribute to a cleaner environment, and promote sustainable waste disposal and recycling efforts. This article also includes various experimental data analyses to demonstrate the suggested strategy’s success, where 98.9% accuracy in garbage sorting has been found using the suggested strategy.

## 8. Patent

Certified the design in patent, government of India named “Garbage collector” and the Design No: 316641-001.

## Figures and Tables

**Figure 1 sensors-23-07963-f001:**
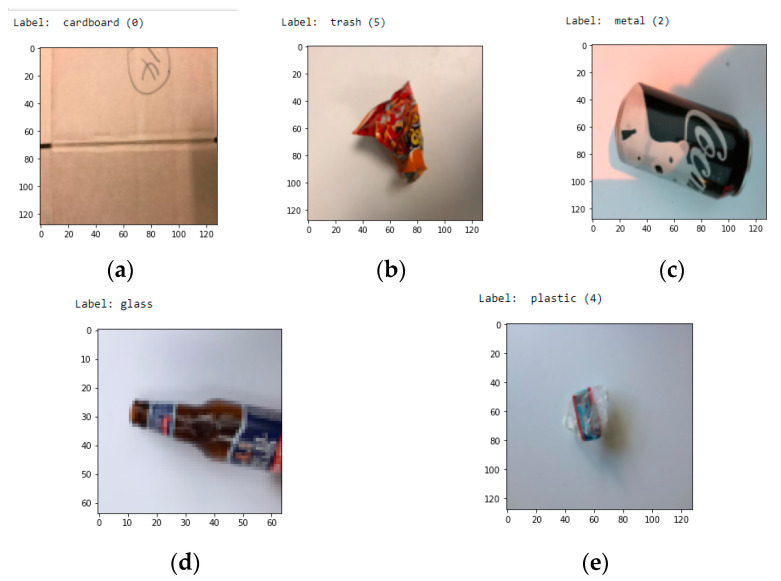
Examples of the input data set for various garbage collection (**a**) cardboard, (**b**) Trash, (**c**) Metal, (**d**) Glass, (**e**) Plastic.

**Figure 2 sensors-23-07963-f002:**
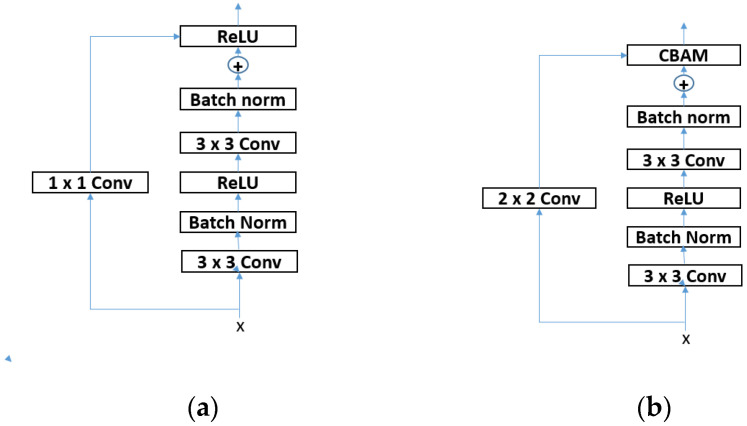
Structure diagram of (**a**) the base residual block of the original ResNeXt and (**b**) modified ResNeXt.

**Figure 3 sensors-23-07963-f003:**
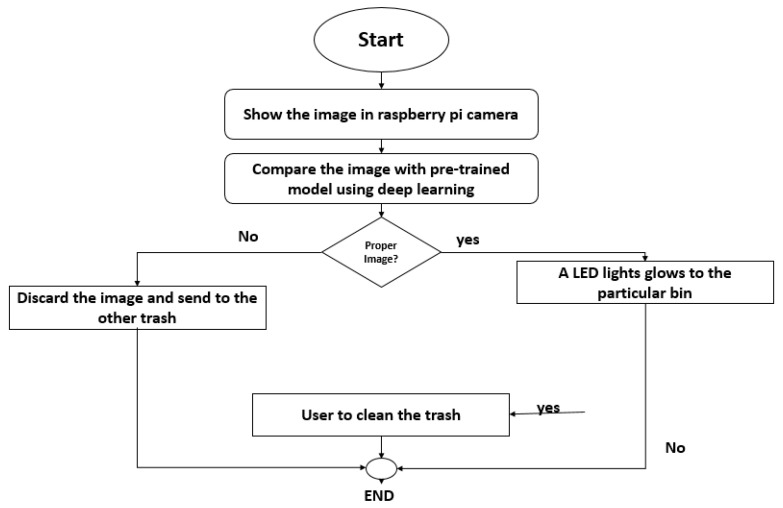
The flow chart of the pi camera.

**Figure 4 sensors-23-07963-f004:**
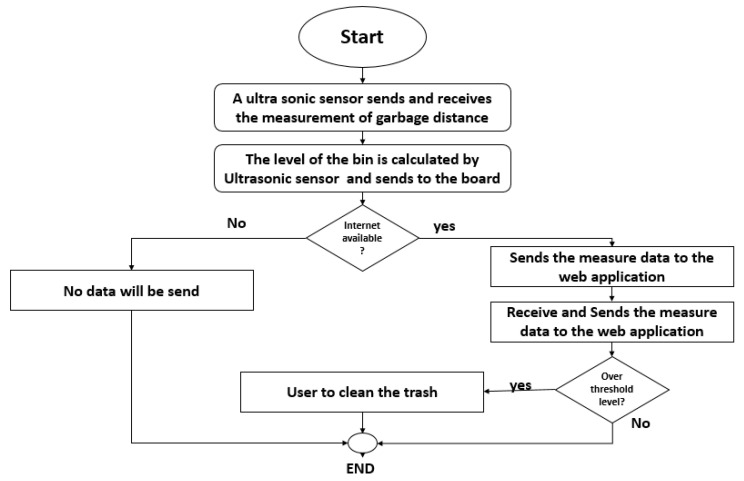
Working principal flow chart of the ultrasonic sensor.

**Figure 5 sensors-23-07963-f005:**
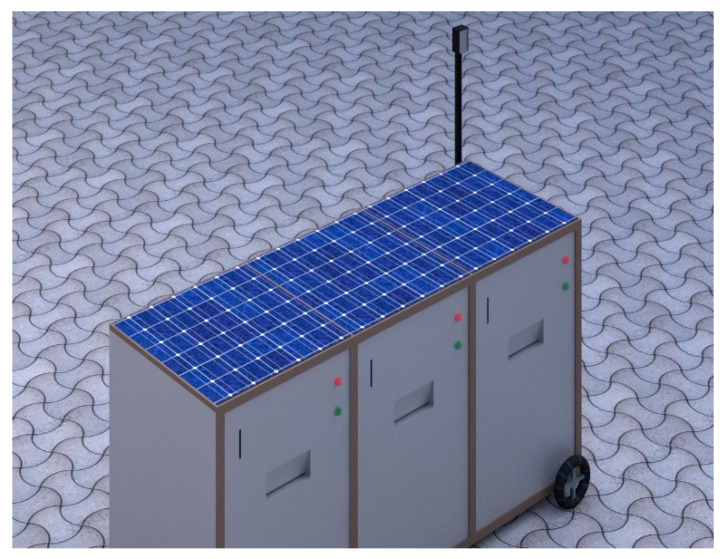
3D view of the proposed model.

**Figure 6 sensors-23-07963-f006:**
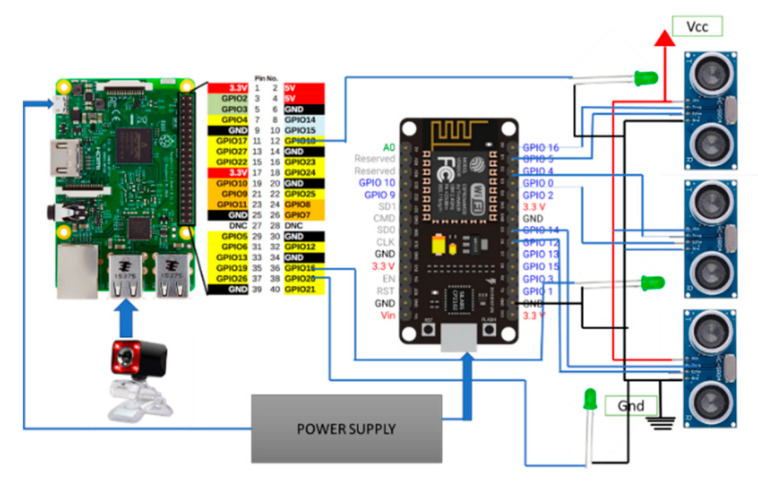
Circuit diagram of smart waste management.

**Figure 7 sensors-23-07963-f007:**
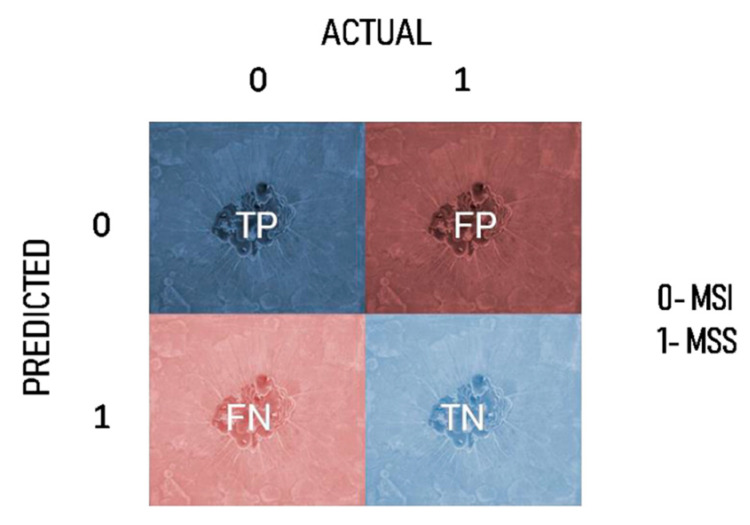
Confusion matrix.

**Figure 8 sensors-23-07963-f008:**
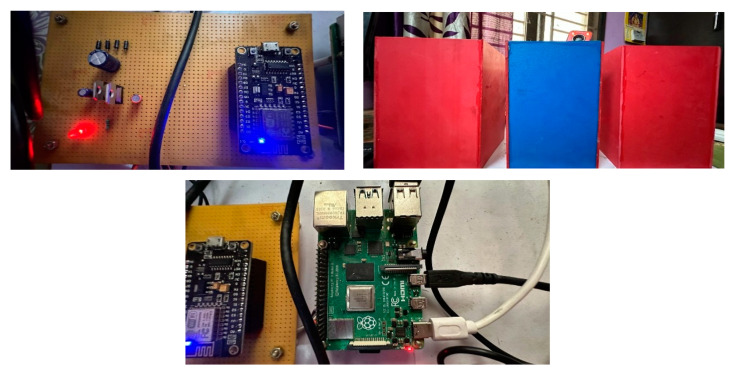
Shows the practical implementation of the proposed work.

**Table 1 sensors-23-07963-t001:** Comparison of various papers.

Paper Title	Authors	Year	Factors Examined	Compromises in Some Areas
A DL method-based hardware solution to categorize garbage in the environment	Tanya Gupta, Rakshit Joshi, Devarshi Mukhopadhyay et al. [33]	2021	The project aims to identify the best-performing CNN model that can be integrated into the hardware-based waste segregation system. By leveraging the power of pre-trained CNNs, the smart bin can accurately categorize different types of waste, enabling efficient waste management, recycling efforts, and a cleaner environment. The findings of the project can have significant implications for improving waste management practices and contributing to sustainable waste disposal strategies.	Minimizing time gaps by cutting back on delays in the system can be a viable approach to improving efficiency, but it should be done with careful consideration of the overall performance and accuracy of the waste segregation system.
Waste classification for sustainable development using image recognition with deep learning neural network models	Meena Malik, Sachin Sharma, Mueen Uddin, Chin Ling Chen, Chih-ming Wu, Punit Soni [34]	2022	The combination of fine-tuning techniques, hardware-based implementation on the Raspberry Pi, and efficient use of computational resources contribute to the success of the suggested method. The 4X improvement in FLOPS efficiency and the ability to create makeshift categories demonstrate the advancements made in waste categorization and smart waste management using this approach.	Implementing the waste categorization model as part of a smart city project comes with various challenges and considerations. Addressing these challenges requires collaborative efforts from stakeholders, including waste management authorities, technology providers, city planners, and the public. By overcoming these issues, the implementation of the waste categorization model as part of a smart city project can contribute to more efficient waste management, reduced environmental impact, and a cleaner and more sustainable urban environment.
Deep Reinforcement Learning Enabled Smart City Recycling Waste Object Classification	Mesfer Al Duhayyim, Taiseer Abdalla Elfadil Eisa, et al. [35]	2022	The suggested method aims to achieve accurate and efficient waste categorization. This method has the potential to significantly enhance waste management practices, as it allows for automated and intelligent waste sorting, promoting recycling and more effective waste disposal strategies. The six waste classes defined by the method provide valuable insights to waste management authorities, enabling better decision-making and resource optimization in waste collection and recycling efforts.	When compared to EfficientNet models, the accuracy in the top-1 is much lower. Furthermore, overfitting and underfitting might occur during the training of the waste category models using a core CNN.
Deep learning-based waste detection in natural and urban environments	Sylwia Majchrowska, Agnieszka Mikołajczyk, Marta A. Plantykow [36]	2022	The study offers a method for locating rubbish in metropolitan environments while taking into account the possibility that the waste may be present indoors, outdoors, or underwater.	The model does not account for the possibility of localizing any object with dimensions that are extremely small in comparison to surrounding objects. Cigarette buts, tiny particles of debris, etc., are examples of this.
Machine Learning and IoT-Based Waste Management Model	Rijwan Khan, Santosh Kumar, Akhilesh Kumar, Srivastava [37]	2021	The hardware prototypes and intelligent waste management strategies presented in the research aim to contribute to a pollution-free and clean environment. By leveraging ML and IoT technologies, waste management can be streamlined, waste segregation can be improved, and appropriate disposal methods can be implemented, ultimately leading to a healthier and more sustainable environment.	The solid waste management architecture, the proposed model, can contribute to improved efficiency, reduced environmental impact, and overall effectiveness in waste management practices. Its integration into the larger waste management system can lead to more sustainable waste management processes and contribute to creating cleaner and healthier environments.
A Waste Classification Method Based on a Multilayer Hybrid Convolution Neural Network	Cuiping Shi, Cong Tan, Tao Wang, Liguo Wang [38]	2021	The suggested methods leverage ML-based image classification techniques to automatically identify and sort trash items into the appropriate waste categories. The use of such advanced technologies helps streamline waste management processes, reduces human effort, and contributes to better environmental protection by promoting recycling and appropriate waste disposal methods.	Additionally, achieving the required accuracy in multiclass categorization is crucial for reliable waste sorting and proper disposal. Further research and development may be necessary to address these limitations and improve the suggested approaches. This could involve exploring more sophisticated machine learning algorithms, increasing the size and diversity of the training dataset, or optimizing the image processing techniques used in waste categorization.
Intelligent waste management system using deep learning with IoT	Md. Wahidur Rahman, Rahabul Islam, Arafat Hasan [39]	2022	The classification accuracy of the proposed architecture, depending on the CNN model, is 95.3125 percent, and its SUS score is 86 percent. However, with real-time waste monitoring, this intelligent system would be adaptable to household activities.	The answer will be more effective, and the accuracy of the plan will also be increased. Future iterations of this research will address these three issues to guarantee even better outcomes in the area of waste management.

**Table 2 sensors-23-07963-t002:** Represents the dataset information.

Class Label Sub-Label	Biodegradable			Non-Biodegradable				Total Images
	Organic Waste	Cardboard	Paper	Metal	Glass	Plastic	Other Trash	
No of images	2150	18,405	1859	1700	1900	2500	1000	10,314

**Table 3 sensors-23-07963-t003:** The accuracy of the proposed model.

Class	Number of Test Images	Accuracy (%)
Cardboard	78	99.72%
Glass	64	97.93%
Metal	53	98.44%
Paper	56	98.58%
Plastic	48	99.63%
Trash	20	98.97%

**Table 4 sensors-23-07963-t004:** Describes the outcomes of various algorithms.

CNN Model	20 Epochs Error	Time (s)	30 Epoch Error	Time (s)
ResNet 50	0.05312	1196	0.06489	2153
Vgg 16	0.07469	1220	0.06897	2384
AlexNet	0.3187	998	0.18695	1569
Modified ResNeXt	0.2156	875	0.15689	1028
ResNet 50 and Modified ResNeXt	0.20107	856	0.14368	986

**Table 5 sensors-23-07963-t005:** Cost occurred for testing and the proposed method.

S.no	Component	Type	Quantity
1	Camera	Low sensitive camera	1
2	Ultrasonic sensor		6
3	Temperature sensor		6
4	Bin	Plastic	1
5	Circuit board	Raspberry Pi 3B	1
6	Battery	Low	1
**Proposed Method**			
1	Camera	Pi camera	1
2	Bin	Stainless steel	
3	Solar panel		
4	Circuit board	Raspberry Pi 3B	1
5	Ultrasonic sensor		6
6	Temperature sensor		6

**Table 6 sensors-23-07963-t006:** Comparison results from various algorithms of the confusion matrix.

Model	True Positive (TP)	False Positive (FP)	False Negative (FN)	True Negative (TN)
Convolution Neural Network	5880	1625	1855	9873
VGG16	0	7505	0	11,728
ResNet 50	6164	1341	972	10,756
ResNeXt	5940	1565	950	10,778
Modified ResNeXt	6329	1153	789	10,785
Modified ResNeXt (Proposed Model)	6338	1167	792	10,936

**Table 7 sensors-23-07963-t007:** Represents the real-time analysis of the proposed garbage bin.

	Time Delay (min)	Waste Level (cm)	Empty Level (%)	Weight of the Waste (kg)
1	1.0	5.00	Above 90%	0.500
2	2.5	9.32	Above 89%	0.647
3	5.5	17.4	Above 70%	0.879
4	10.0	29.0	Below 20%	1.999

## Data Availability

Data are available through free distribution license from Kaggle.

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
