# Peer review of "A Design and Implementation Using an Innovative Deep-Learning Algorithm for Garbage Segregation"

_sensors, 2023, doi:10.3390/s23187963_

Round 1

Reviewer 1 Report

Overall, the article presents a promising solution for waste management using smart garbage bins and deep learning techniques. However, refining the organization, providing more context and analysis of the results, and addressing practical considerations would further strengthen the article's overall impact and appeal to a broader audience.

The abstract of the article provides a brief overview of the waste management challenges posed by lifestyle changes, consumerism, and packaging advancements. It introduces the concept of a three-compactor smart bin system designed to address the problem. It could be improved by providing specific data or statistics on the current waste management issues, the limitations of existing technologies, and how the proposed three-compactor smart bin system overcomes those limitations.

The introduction lacks a clear connection to the proposed three-compactor smart bin system, which is the main focus of the article. The introduction should establish a stronger link between the waste management challenges highlighted and how the innovative smart bin system addresses these challenges.

The article should have provided a discussion on how the proposed smart bin system differentiates itself from the related works and how it addresses some of the limitations or shortcomings observed in previous research.

The conclusion should succinctly summarize the key findings and contributions of the research. It should emphasize the impact of the proposed smart garbage bin system on waste management practices and its potential to address environmental challenges effectively.

Author Response

1. I have included in the table comparision of various algorithm vs proposed algorithm used in the smart bin .Please see the attachment .

Reviewer 2 Report

Authors proposed a DL algorithm for Garbage Segregation. The major issues are listed below.

1. Authors claimed that they have suggested an innovative Deep Learning algorithm for garbage segregation. But we didn't find any DL model/algorithm throughout the manuscript. 

2. The DL model, training and testing metrics are not found in the paper.

3. In the discussion and conclusion, they have claimed that confusion matrix shows that it identified and segregated correctly. But their is no evident in the manuscript.

4.  The related works are not properly discussed and also the problem statement is not clear.

5.   Its is looks like a normal project on garbage segregation and claimed that it segregates using DL algorithm.

Extensive english editing is required.

Author Response

Dear ,

I have attached document.

Thank you 

Round 2

Reviewer 2 Report

Authors addressed all my previous concerns and it can be accepted in the present form.